# Eco-Friendly Activation of Silicone Surfaces and Antimicrobial Coating with Chitosan Biopolymer

**DOI:** 10.3390/ijms262412084

**Published:** 2025-12-16

**Authors:** Daniel Amani, Guðný E. Baldvinsdóttir, Vivien Nagy, Freygardur Thorsteinsson, Már Másson

**Affiliations:** 1Faculty of Pharmaceutical Sciences, School of Health Sciences, University of Iceland, Hagi, Hofsvallagata 53, 107 Reykjavík, Iceland; ama52@hi.is (D.A.); gudnyeyglo@gmail.com (G.E.B.); 2Minamo, Hofsvallagata 53, 107 Reykjavík, Iceland; vivien@hi.is; 3Össur Iceland ehf., Grjóthálsi 5, 110 Reykjavík, Iceland; fthorsteinsson@ossur.com

**Keywords:** silicone, ethanolamine, tensile strength, chitosan, antibacterial activity

## Abstract

Silicone is widely used in medical devices due to its mechanical properties and biocompatibility; however, microbial contamination of silicone surfaces, which can lead to nosocomial infections, remains a significant concern. This can be countered by surface modification using techniques commonly involving oxidative plasma activation or ozone treatments, followed by treatment with silanization agents. Here, we report an alternative surface modification procedure involving treatment with non-toxic organic hydroxyl amines or diamine dissolved in eco-friendly solvents, thus avoiding using reactive and potentially harmful compounds and not requiring specialized equipment. Our findings demonstrate that ethanolamine in isopropanol effectively activates silicone without compromising its tensile strength, making it ideal for further modification. The activated surfaces showed stable amino group areal concentrations over a 10-day period, confirmed by fluorescence imaging and ninhydrin assays. Subsequent treatments with glutaraldehyde and chitosan enhanced the antibacterial properties of the silicone. Chitosan-coated silicone significantly reduced Gram-positive and Gram-negative bacteria colony-forming units (CFUs), with *Enterococcus faecalis* CFUs decreasing from 7.1 to 3.7 Log_10_ CFU/mL. This study introduces a sustainable activation technique for silicone surfaces, resulting in medical devices with improved resistance to microbial colonization while maintaining their mechanical integrity.

## 1. Introduction

Silicones, based on polysiloxane polymers, most commonly polydimethylsiloxane (PDMS), are esteemed for their beneficial properties, making them favored materials across numerous industries. Silicone is widely used in various medical devices, including urinary catheters [1], wound dressings [2], breast implants [3,4], surgical drains [5], gastrostomy tubes [6], cochlear implants [7], ophthalmic devices, and prosthetics [8], due to its excellent biocompatibility, flexibility, and resistance to high and low temperatures [5]. Their flexibility and elasticity allow them to adapt to the body’s shapes, providing comfort and durability in various medical contexts [9]. There are numerous benefits associated with the use of silicone in healthcare applications. However, silicone’s inherent properties, including its hydrophobic surface, also make it a favorable substrate that promotes microbial adhesion and colonization, leading to persistent biofilm formation which poses significant clinical challenges [10]. Bacterial contamination of silicone surfaces is a significant cause of nosocomial infections, particularly in devices such as urinary catheters [11], central venous catheters (CVCs) [12], and silicone-based implants [13]. These infections, often associated with biofilm formation, can lead to severe complications, including bloodstream infections and prolonged hospital stays. For instance, it is estimated that catheter-associated urinary tract infections (CAUTIs) account for up to 40% of all healthcare-associated infections [14,15]. These infections not only increase patient morbidity and mortality but also contribute substantially to healthcare costs, with an estimated annual cost of over 451 million USD annually in the United States alone [16]. Biofilm-associated infections on silicone medical devices are predominantly caused by biofilm-forming bacteria, including *Staphylococcus aureus* (*S. aureus*), *Pseudomonas aeruginosa* (*P.aeruginosa*), *Escherichia coli* (*E. coli*), *Enterococcus faecalis* (*E. faecalis*), *Staphylococcus epidermidis*, *Klebsiella pneumoniae*, and *Proteus mirabilis* [17,18,19]. The specific pathogens may vary depending on the type of the device and its site of application. For example, *Staphylococcus aureus* and *Staphylococcus epidermidis* are the most frequently isolated bacteria in prosthetic and implantable device-related infections, causing approximately 40–50% of prosthetic heart valve infections, 50–70% of catheter biofilm infections, and 87% of bloodstream infections [19]. On the other hand, *E. coli* is the most prevalent pathogen in CAUTIs [15]. Various approaches have been developed to address the challenge of bacterial adhesion and biofilm formation on medical devices. These strategies range from simple antimicrobial treatments to advanced, stable coatings. One of the simplest approaches to enhancing the antibacterial properties of silicone surfaces involves treatment with antimicrobial agents. In this method, silicone surfaces are typically dipped into or sprayed with solutions containing antimicrobial compounds. This approach is cost-effective and straightforward, offering immediate antimicrobial activity. However, the effects are often transient, as the applied agents are susceptible to being removed through washing or degraded during prolonged use, leading to diminished efficacy over time [20].

Silver nanoparticles offer a promising approach for enhancing the antibacterial properties of silicone, particularly in medical applications. They provide broad-spectrum activity against a wide range of pathogens, and their multifaceted mechanisms of action reduce the likelihood of bacterial resistance [21]. However, their clinical applications are limited by potential toxicity, which may adversely affect host cells during antimicrobial treatments [22]. Furthermore, the use of silver-containing dressings for burn wounds has been associated with scar discoloration, as observed in animal models, raising concerns about long-term cosmetic outcomes [23]. Similarly, silver coatings on CVCs have shown mixed results regarding thrombus formation [24].

Coating silicone surfaces with biopolymers through covalent binding offers a promising strategy to address the limitations associated with traditional antimicrobial treatments. These biopolymer-based coatings can exhibit broad-spectrum activity against a wide range of pathogens and be designed to form a stable and durable surface that enhances antimicrobial efficacy while minimizing toxicity to host tissues. For example, Pinese et al. [25] demonstrated the immobilization of hybrid antibacterial peptides on silicone catheters via covalent siloxane bonds, creating a long-lasting and effective antimicrobial barrier. The covalent grafting method avoided toxic reagents, ensuring biocompatibility and scalability for industrial applications. The peptide-grafted silicone surfaces significantly reduced bacterial adhesion and biofilm formation while maintaining antimicrobial efficacy even after sterilization. Another innovative approach involves the covalent grafting of crosslinked poly(poly(ethylene glycol) dimethacrylate) (P(PEGDMA)) layers on silicone surfaces, as demonstrated by Li et al. [26]. This method highlights the potential of polymer-based surface modifications to address challenges in antibacterial and antifouling applications. The incorporation of polysulfobetaine (P(DMAPS)) onto the P(PEGDMA) layer further enhances antifouling and hemocompatibility properties by reducing protein and platelet adhesion. Recently, Cheng et al. [27] explored the potential of hydrogel coatings to address the limitations of silicone rubber in medical applications, particularly its susceptibility to biofouling and microbial infections. They developed a poly(AAm-MPC-ZMA) hydrogel coating, combining acrylamide (AAm), 2-methacryloxyethyl phosphorylcholine (MPC), and zinc methacrylate (ZMA), to enhance the hydrophilicity and antifouling properties of silicone rubber. This coating effectively prevents bacterial adhesion and biofilm formation while enhancing silicone’s biological compatibility and maintaining its protective properties under varied conditions.

Given recent advancements, coating silicone with chitosan presents a particularly promising strategy. Chitosan, a natural biopolymer derived from chitin, is renowned for its broad-spectrum antimicrobial activity against a wide range of bacteria and fungi, as well as its excellent biocompatibility [28,29]. These properties make chitosan-coated silicone surfaces highly effective in preventing pathogen adhesion and biofilm formation, offering a robust solution for various medical applications. Chitosan most likely exerts its antimicrobial effects primarily through electrostatic interactions with negatively charged microbial cell walls. These interactions disrupt membrane integrity, increase cell permeability, and ultimately inhibit microbial growth. This non-specific mode of action significantly reduces the risk of developing bacterial resistance compared to traditional antibiotics, which often target specific biochemical pathways. Additionally, chitosan’s ability to form a protective barrier further enhances the biocompatibility of silicone surfaces [30]. Thus, coating silicone with chitosan not only boosts antibacterial properties but also provides broad-spectrum protection, making it highly suitable for medical devices such as catheters and implants. Chitosan exhibits minimal toxicity and does not provoke significant inflammatory or immune responses, supporting its safe use in long-term medical applications. The combination of chitosan’s wide-ranging antimicrobial efficacy and silicone’s durability offers a compelling approach to preventing infections associated with medical devices.

The covalent binding of antibacterial polymers and compounds to silicone surfaces is challenging due to the inert nature of silicone, which lacks reactive functional groups necessary for modification. Surface activation is typically required to enable the introduction of reactive sites, such as hydroxyl (-OH), silanol (-SiOH), or carboxyl (-COOH) groups, that facilitate the grafting of polymers or biomolecules [31]. The most common procedures involve corona discharge and plasma treatments, creating ozone and other reactive oxygen species that oxidize and replace the methyl groups on the silicone surface with silanol groups (-SiOH), increasing surface polarity and hydrophilicity [32,33]. These treatments significantly enhance the surface energy and wettability of silicone materials, promoting better adhesion of inks, coatings, and adhesives [34]. However, the induced hydrophilicity is often unstable, as silicone surfaces tend to regain their hydrophobicity within hours to days due to the migration of polar groups into the bulk polymer matrix, surface contamination or degradation, and the natural tendency of the polymer to return to a thermodynamically favorable low-energy state [35]. To address this limitation, further chemical treatment, such as silanization, is necessary to stabilize the activated surface and maintain long-term hydrophilicity, particularly in medical device applications.

Silanization involves reacting the silanol groups on the surface with organofunctional alkoxysilane compounds and enabling the covalent grafting of biomolecules or synthetic polymers, preventing the return to a hydrophobic state. Silanization offers the advantage of creating a more durable surface modification, providing long-term stability and precise control over surface chemical functionality [36,37]. While these surface activation methods are effective, they present several practical challenges in the modification of silicone medical devices. Techniques such as plasma treatment require specialized equipment, which may limit their scalability, particularly for devices with complex geometries. Additionally, the generation of ozone and other reactive oxygen species during silicone modification necessitates strict safety measures to minimize potential health risks associated with human exposure [38]. The subsequent silanization step also typically involves using volatile organic solvents, further complicating the process by introducing additional safety and environmental concerns.

In search of a more practical approach with a better safety profile, we found that silicone surfaces can also be activated by treatment with biocompatible hydroxyl amines and diamines. This method provides a simpler, cost-effective, eco-friendly alternative for activating and antimicrobial coating of silicone surfaces. This approach for silicone activation eliminates the need for specialized or high-cost equipment, such as plasma systems, and avoids the use of hazardous or toxic chemicals often employed in some conventional activation techniques. Therefore, the objective of this study was to develop a novel method utilizing biocompatible amine alcohols to achieve stable surface activation without compromising the mechanical properties of the silicone substrate, facilitating the attachment of chitosan as an antimicrobial agent. We specifically aimed to leverage the inherent antimicrobial properties of chitosan to enhance the resistance of silicone-based medical devices against a broad spectrum of bacteria, including both Gram-positive and Gram-negative strains. This technique may address some of the limitations of conventional surface modification methods, offering a safer and more accessible solution for improving the biocompatibility and antimicrobial properties of silicone materials in medical applications.

## 2. Results and Discussion

### 2.1. Surface Activation of Silicone by Reaction with Ethanolamine (ETA) or 1,3-Diaminopropane (DAP)

Silicon (Si) constitutes 28% of the Earth’s crust [39] and is the second most common element after oxygen. These two elements are primarily found in the form of silica (SiO_2_) due to the stability and high bond energy of the Si-O bond (477–549 kJ/mol) [39], and this is a significant factor that contributes to the high thermal and chemical stability of PDMS ([Si(CH_3_)_2_O]_n_) and other silicones. However, this bond can be subject to nucleophilic substitution, and thus, potassium hydroxide in diethylamine has been used for depolymerizing and recycling PDMS to yield cyclosiloxane monomers [40,41,42]. It has also been reported that depolymerization of silicone can also occur to some extent by extensive treatment with diethylamine solvent [43]. Additionally, it has also been shown that the Si-O bonds in organosilanes can be replaced by less stable Si-N bonds by treatment with lithium amides [44].

In pursuit of an environmentally friendly alternative for silicone surface activation, we explored the treatment of silicone elastomers with hydroxyl amine and diamines. Surprisingly, we found that this mild modification approach effectively introduced amino groups onto the surface, which remained stable even after washing. The presence of these amino functionalities was confirmed using ninhydrin assay, demonstrating the efficacy and durability of the surface modification. Figure 1 presents silicone discs treated with 10% ETA in toluene before (Figure 1(aa)) and after ninhydrin treatment (Figure 1(ab,ac), cross-section). Similarly, Figure 1(ba–bc) illustrate the results of treatment with 10% ETA in isopropanol, while Figure 1(ca,cb) depict control samples treated only with the respective solvents. The overall appearance of the silicone material remained unchanged; however, ETA-treated samples exhibited a strong reaction to the ninhydrin reagent, turning dark blue, and indicating the presence of amino groups. In contrast, the control samples did not develop any coloration, indicating the absence of a detectable reaction with ninhydrin. The cross-sectional view further confirms that the color change is confined to the silicone surface, suggesting that the modification is surface-localized rather than penetrating deeply into the material. DAP in both toluene and isopropanol also resulted in surface activation, as indicated by the positive ninhydrin reaction (Figure 1(da–ec)). However, silicone samples treated with DAP in toluene exhibited a distinct change in appearance, transitioning from translucent gray to white, with visible signs of degradation and deformation in shape (Figure 1(da)). The altered morphology and color suggest significant structural modifications, likely due to extensive crosslinking and polymer chain scission facilitated by the swelling effect of toluene.

Compression test results presented in Figure 1(fa–fc) demonstrate the peak force needed to compress the silicone samples by 5 mm. A higher peak force indicates increased stiffness and reduced compressibility, while a lower peak force suggests a softer, more compressible material. When the silicone discs were treated with isopropanol, no significant changes in mechanical properties were observed compared to the control samples. Treatments with DAP or ETA in isopropanol resulted in a slight increase in peak compressive force (Figure 1(fa,fb)), indicating that both compounds induced surface modifications while preventing excessive crosslinking or degradation. This minor increase suggests a subtle reduction in compressibility, although the silicone’s mechanical integrity and elasticity were largely preserved. In contrast, treatment with DAP in toluene led to a significant increase in peak compressive force (Figure 1(fc)), reflecting strong crosslinking effects and a marked decrease in the material’s flexibility and compressibility. Toluene, being a swelling solvent, facilitated deeper penetration of DAP into the silicone matrix, promoting structural modifications that compromised the flexibility and compressibility of the elastomer. On the other hand, isopropanol, a non-swelling solvent, restricted the interaction to the surface, allowing for controlled modification without affecting the bulk properties of the silicone.

### 2.2. Silicone Modification with DAP, ETA, 3-Amino-1,2-Propanediol (APD), and Ethylenediamine (EDA)

Exploratory investigations showed that other solvents such as dimethyl sulfoxide (DMSO), *N*,*N*-dimethylformamide (DMF), water and ethanol could be used for the activation step, with toluene emerging as the most effective medium (Appendix A). Isopropanol facilitated moderate surface activation, whereas water consistently produced the lowest activation levels across all tested amine agents. Specifically, ETA demonstrated the highest surface activation, with a recorded value of 644 ± 30 nanomoles/cm^2^ in toluene, which decreased to 303 ± 31 nanomoles/cm^2^ in isopropanol and further to 62 ± 2 nanomoles/cm^2^ in water (Table 1). The superior performance of toluene can be attributed to its ability to penetrate the silicone matrix, potentially increasing the available surface area for reaction. This effect was evidenced by a notable swelling of the silicone discs to an increased size when disc treated with toluene (Figure 2A). However, after subsequent washing with isopropanol and water, the disc size returned to its original dimensions (Appendix A). These findings also highlight the significant influence of the choice of amine agent in optimizing the surface activation of silicone, with ETA proving to be more effective in silicone activation than other amine agents, particularly in toluene and isopropanol (Table 1).

The experimental data demonstrated a positive correlation between ETA concentration and the number of amino groups on the silicone surface. For instance, when the ETA concentration was raised from 0.25% to 10% (*v*/*v*), the quantity of amino groups on the silicone surface increased from 459 to 4497 nanomoles/cm^2^ in toluene. A similar trend was observed when reactions were conducted in both water and isopropanol (Figure 2B). However, increasing the concentration above 10% (*v*/*v*) did not lead to a further increase in amino groups on the surface. Light microscopy images of silicone discs treated with ETA, followed by the ninhydrin assay, indicated that the reaction between ETA and the silicone was limited to the surface of the discs (Figure 2C).

### 2.3. Stability of the Surface Activation

The stability assay aimed to investigate the resistance of the surface-bound amino groups on activated silicone to removal by washing with deionized water. The ninhydrin assay revealed that amino group retention decreased significantly within the first 24 h, dropping from 366 nanomole/cm^2^ to 206 nanomole/cm^2^ (about 56% of the initial value). After two days, the amount was 180 nanomole/cm^2^, and by day 4, it stabilized at around 161 nanomole/cm^2^ (44% of the initial value). From day 4 through day 10, retention remained steady, ranging from 160–165 nanomole/cm^2^ (Figure 3A).

The discs were treated with amino reactive fluorescein isothiocyanate (FITC) in order to further confirm the activation and to evaluate the stability of the coating. Fluorescence imaging of control samples demonstrated that silicone surfaces that had not been subject to pre-treatment exhibited negligible fluorescence, indicating minimal non-specific adsorption of FITC. In contrast, ETA-activated silicone discs showed strong green fluorescence, verifying the successful covalent binding of the isothiocyanate groups of FITC and the amino groups on the surface of the silicone. After incubation in water for 10 days, the fluorescence intensity remained stable (Figure 3B).

### 2.4. Chitosan Coating of ETA-Activated Silicone via Glutaraldehyde Crosslinking

Activated silicone discs were coated with chitosan and investigated for antimicrobial activity against *S. aureus*, *E. faecalis*, *E. coli*, and *P. aeruginosa*, which commonly cause hospital-acquired infections associated with silicone-based medical devices. The ETA-coated silicone discs were treated with chitosan using glutaraldehyde as a crosslinker to facilitate covalent bonding between the amine groups of chitosan and ETA (Figure 1).

The crosslinking effect of glutaraldehyde on silicone/ETA discs was quantitatively monitored using the ninhydrin assay to measure surface amino groups. After 30 min of glutaraldehyde treatment, the amino content significantly decreased to 112 ± 5 nanomoles/cm^2^. This reduction continued, with levels dropping to 4.9 nanomoles/cm^2^ at 90 min. Beyond this point, no substantial changes were observed up to 180 min, indicating that most reactive amino groups were crosslinked within the first 90 min (Figure 4A). Following the glutaraldehyde reaction, the ninhydrin assay could no longer detect amino groups, as the free NH_2_ groups had formed covalent bonds, rendering them unreactive with ninhydrin. This observation indicates that the surface modification was successful, with a thorough crosslinking process that enhances the stability of the silicone surface for future functionalization.

The impact of chitosan concentration and incubation time on the quantity of amino groups on ETA/glutaraldehyde-treated silicone was examined via ninhydrin assay in this study. For instance, after a 30-min incubation, increasing the concentration of chitosan from 1% to 3% resulted in a substantial rise in the surface concentration of amino groups, from 94 ± 3 nanomoles/cm^2^ to 853 ± 8 nanomoles/cm^2^ (Figure 4B). A similar trend of an increase in amino groups with rising chitosan concentration was observed across different treatment durations. The increase in amino groups after reacting chitosan with glutaraldehyde-treated silicone is due to the additional amino groups contributed by the chitosan, which contains multiple amino functional groups. As chitosan concentration increases, more amino groups are introduced onto the surface, resulting in a higher detectable concentration. Conversely, extending the incubation time did not significantly influence the amino group concentration, indicating saturation of the binding after less than 15 min treatment with chitosan solution.

Figure 5 displays a comparative analysis of the FTIR spectra for silicone, chitosan, and chitosan-coated silicone. The FTIR analysis highlights similar peaks between pure chitosan and chitosan-coated silicone, indicating the presence of chitosan on the silicone surface. Pure chitosan exhibited characteristic absorption peaks around 3360 cm^−1^ for N-H and O-H stretching, 2877 cm^−1^ for C-H stretching, 1585 cm^−1^ for N-H bending (amide II), and 1371 cm^−1^ for C-N stretching [45]. Similarly, the chitosan-coated silicone exhibited peaks at 3261 cm^−1^ (N-H and O-H stretching), 2962 cm^−1^ (C-H stretching), 1551 cm^−1^ (N-H bending), and 1404 cm^−1^ (C-N stretching). In addition to the chitosan peaks, the chitosan-coated silicone spectrum also displays signature peaks characteristic of silicone, including the Si-O-Si stretching vibration at 1011 cm^−1^, and the Si-CH_3_ stretching at 1259 cm^−1^ [46]. The presence of these peaks in the chitosan-coated silicone spectrum suggests successful coating of chitosan on the silicone surface. The slight shifts in peak positions or intensities in the coated samples compared to pure chitosan can be attributed to the interaction between chitosan and the silicone surface, confirming the successful coating of silicone with chitosan.

The hydrophilicity and surface morphology of the silicone samples were significantly altered by ethanolamine (ETA) activation followed by chitosan coating. Although ETA-treated silicone exhibited a higher water contact angle than untreated silicone, the contact angle decreased substantially after chitosan coating, indicating enhanced surface hydrophilicity (Appendix A). SEM analysis revealed that untreated silicone displayed a porous and heterogeneous morphology, whereas ETA-activated silicone exhibited a smoother surface. Chitosan-coated silicone showed distinct microstructural patterns at higher magnifications, confirming the presence of a continuous coating layer (Appendix A).

Biocompatibility and long-term stability of chitosan coating on silicone surfaces under conditions relevant to medical applications are key factors in ensuring the efficiency and safety of this method. Chitosan, as a natural biopolymer, has excellent biocompatibility and low toxicity. It can remain in the body for months before gradually degrading, with its degradation products being harmless and excreted without causing tissue damage or significant inflammatory responses. These properties indicate that chitosan coatings can be stably and safely maintained in environments associated with the human body, making them suitable for use in medical devices. Although the stability of activated groups on the silicone surface shows considerable persistence in polar environments after the initial stabilization, the durability of the chitosan coating attached to these groups in the body depends on various factors such as the degree of deacetylation, molecular weight, the extent and type of cross-linking, biological conditions at the site of application, and the activity of degradative enzymes. Further studies are needed for a more precise assessment of the degradation and absorption of these coatings under long-term clinical conditions.

### 2.5. Antibacterial Properties

The antibacterial activities of chitosan-coated silicone discs were assessed using the drop plate method to evaluate their efficacy against various Gram-positive and Gram-negative bacterial strains. The results showed that chitosan-coated silicone discs exhibited significant antibacterial activity against all tested bacterial strains compared to uncoated silicone (control), and steel discs, while copper discs (positive control) completely inhibited bacterial growth, showing no colonies present (Appendix A).

Specifically, in the case of *E. faecalis*, a significant reduction in colony-forming units (CFUs) was observed on silicone discs coated with chitosan, with CFUs reducing from 7.1 to 3.7 Log_10_ CFU/mL after the silicone coating process. Furthermore, the presence of chitosan on the silicone surface significantly reduced the CFUs of *S. aureus* and *E. coli*, from 6.3 and 6.5 to 4.9 and 4.6 Log_10_ CFU/mL, respectively, illustrating the coating’s broad-spectrum antibacterial action (Figure 6). In this study, steel and copper discs were selected as controls to create a baseline for comparison, where steel represented a non-antibacterial negative control, and copper was used as a positive control due to its well-known antibacterial capabilities [47].

Steel discs exhibited CFU counts of 5.9, 5.2, 4.8, and 4.1 Log_10_ CFU/mL against *S. aureus*, *E. faecalis*, *E. coli*, and *P. aeruginosa*, respectively, which were higher than the counts on chitosan-coated silicone for the same bacterial strains. Copper discs exhibited no bacterial growth across all tested strains, indicating their superior antibacterial activity. The results of our study highlight the antibacterial activity of chitosan-coated silicones against both Gram-positive and Gram-negative bacteria classes.

In this study, the antibacterial efficacy of chitosan-coated silicone was evaluated against four clinically relevant bacterial strains. We acknowledge that expanding testing to include additional Gram-positive, Gram-negative, and fungal strains would offer a more comprehensive assessment of the coating’s antimicrobial spectrum. To address this, we have obtained fungal strains including *Aspergillus niger*, *Penicillium pinophilum*, *Paecilomyces variotii*, *Trichoderma virens*, and *Chaetomium globosum*, which are commonly used in antifungal studies. Antifungal testing of the chitosan-coated silicone against these strains is planned as part of future work. While quantification of fungal growth on silicone surfaces is challenging despite adherence to ISO 846 standard protocols [48], we are optimizing complementary molecular and staining-based assays to improve sensitivity and accuracy. These efforts will facilitate a robust evaluation of the coating’s broad antimicrobial potential in ongoing and subsequent research.

### 2.6. Impact of Amine Agents and Reaction Medium on the Tensile Properties of Silicone

The impact of amine agents with high effectiveness on silicone activation, including ETA and DAP on the mechanical integrity of silicone was investigated using a texture analyzer. Furthermore, the silicone structure underwent monitoring throughout post-activation processes, such as crosslinking with glutaraldehyde and coating with chitosan, to ensure the mechanical properties of the final antimicrobial silicone product remained unchanged. The results indicate that the tensile properties of silicone can be significantly influenced by the choice of amine agent and reaction medium.

The use of toluene as a reaction medium remarkably changes the tensile properties for both ETA and DAP, suggesting a strong interaction between the silicone matrix and the treatment agent that changes the material’s mechanical properties.

Silicone treated with ETA using isopropanol as a reaction medium showed no significant change in elongation and maximum stress, with values of 318 ± 15% for elongation and 2.3 ± 0.08 MPa for maximum stress compared to the untreated silicone (control: 320 ± 5% elongation, 2.2 ± 0.1 MPa maximum stress). In contrast, most of the tensile properties changed significantly after treating silicone with ETA in toluene as reaction media. For example, the maximum stress and the Young’s modulus increased to 3.8 ± 0.08 MPa and 0.929 ± 0.86 MPa, respectively (Toluene/ETA). However, the elongation at break decreased substantially from 320 ± 5% (control) to 236 ± 26% for Toluene/ETA (Figure 7). These changes demonstrate a substantial alteration in the mechanical properties of the silicone.

## 3. Materials and Methods

### 3.1. Materials

Reagent-grade chemicals, including FITC, glutaraldehyde, lithium acetate dihydrate, sodium borohydride (NaBH_4_), sodium phosphate dibasic dodecahydrate, glacial acetic acid, Mueller-Hinton broth and Mueller-Hinton agar were purchased from Sigma Aldrich (Schnelldorf, Germany). Solvents such as methanol, acetone, isopropanol, and toluene were also purchased from Sigma Aldrich (Schnelldorf, Germany). Chitosan with a molecular weight of 108 kDa and a degree of acetylation (DA%) of 83% was kindly provided by Primex ehf (Siglufjordur, Iceland). Ninhydrin, hydrindantin dihydrate, DMSO, and all amine agents, including 1,3-diaminopropane, ethanolamine, 3-amino-1,2-propanediol, 3-amino-1-propanol, and ethylenediamine, were purchased from Tokyo Chemical Industry Co., Ltd. (Zwijndrecht, Belgium). Sodium chloride was obtained from Merck (Darmstadt, Germany), and potassium dihydrogen phosphate was supplied by Fluorochem EU Limited (Cork, Ireland).

### 3.2. Methods and Preparations

#### 3.2.1. Curing of Silicone Membranes and Sample Disc Preparation

Platinum-cured LSR (Liquid Silicone Rubber) silicone sheet samples were prepared by thoroughly mixing 1:1 ratio of CF13 (Nusil Technology LLC, Carpinteria, CA, USA), parts A and B in a mixer under vacuum and placing the uncured mixture in a homemade sample press and curing at 100 °C for 18 min. The resulting 2 mm-thick sample sheet was then removed from the sample press and cut to the desired sample size.

The experimental procedure started with the preparation of uniform 4 mm silicone discs with 2 mm thickness using a precision hole punch. The discs were immersed in acetone and ultrasonicated for 30 min to remove all surface contamination. Subsequently, these discs were washed in 50 mL of water while stirring at 250 rpm for 30 min with a magnetic stirrer, a process that was repeated twice. This was followed by a similar washing sequence in 25 mL of isopropanol. For the treatment phase, separate reaction media were prepared with 20 mL of either water, toluene, or isopropanol, each containing a 10% (*v*/*v*) hydroxyl amine compound or diamine.

The reagents used for activating the silicone were DAP, ETA, APD, 3-amino-1-propanol (3AP), and EDA, prepared individually in each solvent. Silicone discs were immersed in the amine-based solutions and stirred at 250 rpm for 24 h under ambient conditions, with a minimum of three replicates for each treatment. After the reaction period, the supernatants were removed, and the silicone discs were washed with 25 mL of isopropanol under stirring at 250 rpm for 30 min. This was followed by a final wash in 50 mL of deionized water, repeated twice under the same stirring conditions, to ensure the removal of any unreacted amine agents.

#### 3.2.2. Surface Characterization

Static water contact angle measurements of silicone samples were performed to assess changes in hydrophilicity, and surface morphology of the samples were examined by scanning electron microscopy (SEM) (see Appendix A).

#### 3.2.3. Mechanical Testing

Samples of silicone were prepared using a die cutter to produce dog bone-shaped cutouts with specified dimensions of 12 mm in length and 3 mm in width, with a thickness of 2 mm. The tensile properties of silicone samples were determined using a TA.XTplusC Texture Analyser (Stable Micro System, Godalming, UK). Force was measured in newton (N) for the silicone samples. Parameters were set to a distance of 100.0 mm, test speed of 2.0 mm/s, force of 0.981 N, time to 5.00 s, and loaded cell of 5 kg. In addition to the tensile test, a compression test was conducted on cylindrical silicone samples with a diameter of 12 mm and a thickness of 10 mm. The test mode was set to compression, with a pre-test speed of 1.00 mm/s, a test speed of 2.00 mm/s, and a post-test speed of 10.00 mm/s. The target mode was set to distance, with a compression distance of 5.000 mm and a trigger force of 0.049 N. All measurements were conducted with five replications.

#### 3.2.4. Ninhydrin Assay

The efficacy of the surface modification process was evaluated by quantifying the amino groups present on the silicone surface using the ninhydrin assay. The reagent was freshly prepared by dissolving 1.0 g of ninhydrin and 0.15 g of hydrindantin dihydrate in 37.5 mL of dimethyl sulfoxide (DMSO) in a dark, airtight flask. The solution was then deoxygenated by flushing with nitrogen gas to prevent oxidation, and 12.5 mL of 4 M lithium acetate buffer (pH 5.2) was injected into the mixture and stirred at 250 rpm for 10 min [49]. Silicone samples were immersed in 1 mL of deionized water and 1 mL of the freshly prepared ninhydrin reagent and incubated at 80 °C for 60 min to facilitate the reaction. After incubation, the samples were cooled to room temperature, diluted seven-fold with a water/ethanol (50:50) mixture, and absorbance was measured at 570 nm using a Lambda 35 UV-Vis spectrophotometer (PerkinElmer, Waltham, MA, USA). Amine group quantification was performed by constructing a calibration curve with glucosamine hydrochloride as the standard. Glucosamine hydrochloride solutions were prepared in water at different molar concentrations (0.5 mM, 1 mM, 2 mM, and 5 mM). Each solution was reacted with freshly prepared ninhydrin reagent and processed under the same reaction, incubation, dilution, and measurement conditions as described above. Since glucosamine hydrochloride contains a single primary amine group, the molar concentration of glucosamine hydrochloride corresponds directly to the molar concentration of amine groups. Therefore, the absorbance values obtained at 570 nm were plotted against glucosamine concentrations to generate the calibration curve for amine group quantification. The results were expressed as nanomoles of NH_2_ groups per square centimeter of the silicone disc area.

#### 3.2.5. Glutaraldehyde Treatment

ETA-treated silicone discs were submerged in a 25% glutaraldehyde solution in water, which was stirred intermittently. After exposure times of 30 min, 60 min, 1.5 h, 2 h, 2.5 h, and 3 h, the discs were removed and washed several times on each side with deionized water using a transfer pipette to remove unreacted glutaraldehyde. Subsequently, the discs were dried at room temperature, and the crosslinking efficiency was evaluated by calculating the amount of free amine groups on the silicone surface using ninhydrin assay.

#### 3.2.6. Investigation of the Effects of ETA Concentration on Silicone Surface Activation in Various Reaction Media

To investigate the impact of ETA concentration on the activation of silicone surfaces, silicone discs were treated with a series of ETA solutions at concentrations of 0.25, 0.5, 1, 2.5, 5, and 10% (*v*/*v*) in water, isopropanol or toluene. The mixtures were allowed to stir (250 rpm) at room temperature for 48 h. Subsequently, the silicone discs were washed twice with isopropanol and deionized water to remove free ETA, and amino groups on the surface of silicone discs were quantified using the ninhydrin assay as mentioned above. All the measurements were done in triplicates.

#### 3.2.7. Fluorescence Labeling

Fluorescence labeling was performed to confirm the activation of the silicone surface and its stability over time. For this purpose, both control (untreated) silicone discs and those treated with a 10% (*v*/*v*) solution of ETA in isopropanol were submerged in a 20 mL isopropanol solution containing 1 mg of amine-reactive fluorescein isothiocyanate (FITC). The mixture was then stirred at 250 rpm at room temperature for 24 h. Then, the silicone discs were washed twice with isopropanol and deionized water according to the optimized washing protocol described above to remove the free FITC on the surface of the silicone discs. The fluorescence of each sample was evaluated under UV light to determine the presence of the FITC on the silicone surface. To assess its stability, the FITC-labeled discs were immersed in deionized water over a period of 10 days. Then, fluorescence observations were performed at predefined intervals to monitor the persistence of FITC labeling, indicating the stability of the FITC on the silicone surface.

#### 3.2.8. Stability Study

The stability of amino groups on the surface of silicone treated with ETA was assessed by ninhydrin assay. For this purpose, ETA-treated silicone discs were prepared as mentioned above, and then three silicone discs were immersed separately in 20 mL of deionized water. At scheduled times points, the discs were removed, and ninhydrin assays were performed to quantify the remaining amino content on the silicone surface. The percentage of retained amino groups at each time point was calculated by comparing the absorbance to the value obtained for ETA activated silicone with no prior incubation (day 0), which was defined as 100% activation.

#### 3.2.9. Chitosan Coating

Chitosan solutions were prepared at concentrations of 1%, 2%, and 3% by dissolving 0.5, 1, and 1.5 g of chitosan in 50 mL of deionized water containing 2% acetic acid, respectively. The solutions were stirred at 250 rpm for 1 h to achieve homogeneous mixtures. Subsequently, NaBH_4_ was added in amounts of 13 mg, 25 mg, and 38 mg, respectively, followed by stirring for an additional hour to ensure thorough dissolution of all components. ETA/glutaraldehyde-treated silicone discs were submerged in the chitosan solution for various durations of up to 2 h, with occasional stirring. Then, the discs were thoroughly rinsed with deionized water and subsequently dried in an oven at 80 °C for 1 h. Fourier transform infrared (FTIR) spectroscopy was performed by Thermo Scientific™ Nicolet™ iZ10 FTIR instrument (Thermo Fisher Scientific, Hvidovre, Denmark) and ninhydrin assay was utilized to confirm the attachment of chitosan to the silicone surface and to quantify the amino groups on the silicone surface.

#### 3.2.10. Antibacterial Activity

The antibacterial activity of chitosan-coated silicone discs was assessed using discs coated with 3% chitosan through the drop plate method against two Gram-positive bacteria, *S. aureus* (ATCC 25923) and *E. faecalis* (ATCC 29212), as well as two Gram-negative bacteria, *E. coli* (ATCC 25922) and *P. aeruginosa* (ATCC 27853). For this purpose, fresh bacterial suspensions were prepared by culturing single colonies of each bacterium in 3 mL of Mueller-Hinton broth (MH) and incubating them overnight at 37 °C. Then, the bacterial concentration was adjusted to 1.5 × 10^8^ CFU/mL by diluting the culture with phosphate-buffered saline solution (PBS) to achieve equivalence with a 0.5 McFarland standard. Subsequently, 50 µL of the bacterial suspensions were applied to Mueller-Hinton Agar (MHA) plates using a sterile pipette. The suspensions were distributed across the agar surface using a sterilized cotton swab. After that, the silicone, steel, copper and the chitosan-coated silicone discs were placed on the surface of the MHA plates and incubated at 37 °C. After 24 h, the discs were carefully removed from the agar plates and placed in 3 mL of PBS. Subsequently, the samples were vortexed for one minute to detach bacteria that had adhered to the disc surface. Serial dilution was conducted by diluting the suspensions in a 1:9 ratio. Then, 10 µL of the diluted suspension was plated onto MHA in triplicate and incubated at 37 °C. The colonies on the plates were counted 18 h after incubation and the number of CFUs per milliliter of the original bacterial suspension was then calculated by multiplying the counts by the dilution factor.

## 4. Conclusions

This study presents a novel, simple, and environmentally friendly approach for activating silicone surfaces, which can be used for coating silicone-based medical devices. By utilizing amine-based strategies, this method enables effective surface activation while preserving the tensile integrity of the material. The results highlight that the choice of amine agent and reaction medium significantly influences the efficiency of activation, providing a practical and scalable solution for biomedical applications. ETA in toluene and isopropanol showed a significant capacity for silicone surface activation. However, isopropanol was selected as the favored reaction medium for further modifications, such as chitosan coating, due to its minimal impact on the silicone’s tensile properties. Fluorescent imaging of FITC-labeled ETA-modified silicone revealed sharp and stable fluorescence emission over a 10-day period, confirming the sustained presence of ETA on the surface. These findings highlight the effectiveness of ETA in activating the silicone surface in isopropanol, ensuring high stability without changing the mechanical properties of silicone. Our study demonstrated that the application of chitosan coating on silicone significantly reduces bacterial colonization across a range of Gram-positive and Gram-negative bacteria. Given chitosan’s biocompatibility and broad-spectrum antimicrobial properties, its application in clinical settings presents a promising strategy to reduce infections related to medical devices. Future research will focus on evaluating the long-term stability and biocompatibility of the activated silicone, as well as investigating the antimicrobial efficacy against a broader spectrum of bacteria and fungi in both in vitro and in vivo settings to ensure their safety and effectiveness for medical applications.

## Data Availability

All datasets generated or analyzed during this research are included within the article and Appendix A. For any further information or additional materials related to this study, interested readers may contact the corresponding author.

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
