# Peer review of "Eco-Friendly Activation of Silicone Surfaces and Antimicrobial Coating with Chitosan Biopolymer"

_ijms, 2025, doi:10.3390/ijms262412084_

Round 1

Reviewer 1 Report

Comments and Suggestions for Authors

Dear Authors,
I read your paper with interest. It addresses an interesting topic that has potential applications in healthcare and perhaps other fields. The paper's structure is logical and coherent. However, there are a few issues that need clarification.

Figure 1fa - 1fc. Each figure contains several lines in the same color. Does this indicate repeated measurements? If so, please indicate this in the figure caption.

Line 261
To which Figures 2A, 2B, and 2C does the statement ("The values ​​represent the mean standard deviation") refer? Please move this sentence to the appropriate caption.

Line 307-319
Why were chitosan concentrations higher than 3% not tested? The results presented in Figure 4B suggest that higher concentrations should also react.

Section 2.5
What silicone discs were used for antimicrobial testing? Those coated with 1%, 2%, or 3% chitosan? I didn't see this information in the paper.

Mechanical testing.
Tensile strength is expressed in N, and after converting to the cross-sectional area of ​​the sample, the result is given in MPa. In your paper, the results are presented in completely different units. Why?
The results given in Table 2 are incomprehensible to me.
I also doubt whether a tensile rate of 10 mm/s is appropriate for your material (line 431). Isn't this too high?
Was the elongation of the sample measured during tensile testing?

Author Response

Response to Reviewer 1 Comments

1. Summary

Dear Reviewer,

Thank you very much for taking the time to review our article and providing valuable feedback. We have carefully revised the article in accordance with all your comments. In particular, the tensile properties section was revised to clarify units and tensil properties. Detailed and point-by-point responses are provided below, and the corresponding corrections are included in the resubmitted article, where you can see the changed sections highlighted in red.

2. Questions for General Evaluation

Reviewer’s Evaluation

Response and Revisions

Does the introduction provide sufficient background and include all relevant references?

Yes

Are all the cited references relevant to the research?

Yes/Can be improved/Must be improved/Not applicable

Yes, all cited references have been carefully selected for their relevance to the research.

Is the research design appropriate?

Yes

Are the methods adequately described?

Can be improved

There were a few minor errors in the methods that have now been corrected.

Are the results clearly presented?

Can be improved

The results in sections such as the mechanical testing, where there were ambiguities, have been completely rewritten to improve clarity.

Are the conclusions supported by the results?

Yes

Are all figures and tables clear and well-presented?

Must be improved

We have added solvents and reactants directly on Fig. 1 images and replaced Table 2 with a graph for better clarity.

3. Point-by-point response to Comments and Suggestions for Authors

Comments 1: Figure 1fa - 1fc. Each figure contains several lines in the same color. Does this indicate repeated measurements? If so, please indicate this in the figure caption.

Response 1:

Thank you for your comment. Yes, the repeated lines in the same color do indicate repeated measurements. We have clarified this point in the revised figure caption (Page 5; Lines 229-230).

Comments 2: Line 261

To which Figures 2A, 2B, and 2C does the statement ("The values ​​represent the mean standard deviation") refer? Please move this sentence to the appropriate caption.

Response 2: Thank you for your comment. As suggested, the sentence "The values represent the mean ± standard deviation" has been moved and now appears only in the caption for Figure 2B, where it is directly relevant (Page 7; Line 266).

Comments 3: Line 307-319

Why were chitosan concentrations higher than 3% not tested? The results presented in Figure 4B suggest that higher concentrations should also react.

Response 3: Thank you for your comment. Concentrations above 3% were tested during preliminary experiments. However, at higher concentrations, chitosan became difficult to dissolve, and the solution's viscosity increased significantly, which made achieving a uniform disc coating challenging. Additionally, high viscosity hindered the proper immersion of the silicone discs and disrupted the magnetic stirring during the coating process.

Comments 4: Section 2.5

What silicone discs were used for antimicrobial testing? Those coated with 1%, 2%, or 3% chitosan? I didn't see this information in the paper.

Response 4: Thank you for your attention. In this study, antimicrobial assays were performed using silicone discs coated with 3% chitosan. This information has now been included in the revised manuscript (Page 15; Lines 552-553).

Comments 5: Mechanical testing.

Tensile strength is expressed in N, and after converting to the cross-sectional area of ​​the sample, the result is given in MPa. In your paper, the results are presented in completely different units. Why?

The results given in Table 2 are incomprehensible to me.

Was the elongation of the sample measured during tensile testing?

Response 5: Thank you for your comment. Initially, we intended to present the results in terms of Young’s modulus and MPa. However, we were unable to find the option to manually enter the sample dimensions in our device settings during measurement. Since our main objective was to compare the tensile properties of silicone after activation and coating with the control samples, we reported the device’s output values directly in the manuscript. We extracted the raw data as force (in newtons) and distance (in millimeters). These values were then converted to stress (force divided by cross-sectional area) and strain (change in length divided by original length), which were used for calculating Young’s modulus, maximum stress, and elongation. The results are now presented in the revised manuscript in standard units (e.g., % for elongation, MPa for maximum stress and Young’s modulus) (Page 12; Lines 414-421).

Comments 6: I also doubt whether a tensile rate of 10 mm/s is appropriate for your material (line 431). Isn't this too high?

Response 6: Thank you for your attention and careful review. The tensile rate of 10 mm/s was actually used for the post-test phase and was mistakenly indicated as the testing speed. The correct tensile test speed was 2 mm/s. This error has now been corrected in the revised version (Page 13; Line 469).

4. Response to Comments on the Quality of English Language

Reviewer’s Evaluation: The English is fine and does not require any improvement.

Response 1: Thank you for confirming that the language is satisfactory and does not need improvement.

5. Additional clarifications

Reviewer’s comment regarding the quality of figures: Figures and tables must be improved

Response: Table 2 from the previous version of the manuscript has been revised, and its data are now presented in the form of a bar graph in Figure 7 for greater clarity in the revised version of the manuscript

Reviewer 2 Report

Comments and Suggestions for Authors

The analyzed manuscript presents a novel and eco-friendly method for activating silicone surfaces using biocompatible amine-based compounds in non-toxic solvents. The authors provide comprehensive experimental data on surface activation, mechanical properties, stability, and antibacterial performance, highlighting the potential of this strategy for improving silicone-based medical devices. The findings have important implications for the development of silicone-based medical devices with enhanced antimicrobial properties that could help reduce nosocomial infections.

The manuscript is well-structured, with clear sections for methods, results, and discussion. Figures and tables are informative and effectively support the findings. However, to fully demonstrate the advantages and reliability of the proposed method, the study requires additional comparative data, broader antimicrobial evaluation, and preliminary long-term stability results. Therefore, the authors should carefully reconsider the following points:

i) The authors do not provide sufficient characterization of the silicone support. It is unclear whether the substrate is crystalline or amorphous, which could significantly influence the surface activation process and coating efficiency. Clarifying the structural nature and relevant properties of the silicone material is essential for reproducibility and for understanding the observed results.

ii) While the study discusses the limitations of traditional activation techniques, it does not include experimental comparisons with established approaches such as plasma activation or silanization. Incorporating such comparisons would strengthen the case for the proposed eco-friendly method by demonstrating its relative efficiency, stability, and safety.

iii) The authors are encouraged to include direct experimental comparisons between the proposed activation method and conventional techniques, supported by quantitative data.

iv) The authors mention plans for future research but provide no preliminary data or discussion on potential challenges. Including initial stability and biocompatibility results (particularly under conditions mimicking real-world medical use) would enhance the practical relevance of the study.

v) The antibacterial efficacy of chitosan-coated silicone is evaluated against four bacterial strains. Although these are clinically relevant pathogens, expanding the tests to include additional gram-positive, gram-negative, and fungal strains would provide a more comprehensive assessment of the coating’s antimicrobial spectrum. At minimum, a short comment addressing this limitation should be added, with an indication that broader studies could be presented in future work.

vi) A concise discussion on long-term stability and biocompatibility under medically relevant conditions should be included to provide a more complete perspective on the method’s applicability.

vii) The authors disclose potential conflicts of interest related to a patent and a start-up company. While this transparency is commendable, independent validation of the findings would help mitigate concerns about bias and strengthen the study’s credibility.

I recommend a major revision before the manuscript can be considered for publication.

Author Response

Response to Reviewer 2 Comments

1. Summary

Dear Reviewer,

Thank you for your suggestions and comments, which have helped us improve the quality and clarity of our manuscript. We appreciate your concerns regarding the lack of direct experimental comparisons with conventional activation methods, and the need for preliminary data on stability and biocompatibility. While experimental comparisons with plasma activation and silanization were beyond the scope of this study due to technical limitations and resource constraints, we discussed the rationale and highlighted these limitations transparently. We also included a concise discussion of preliminary stability and biocompatibility data and outlined ongoing experiments that will be detailed in future work.

We also tried to address your concerns on the scope of antibacterial efficacy testing by explaining our current and planned antifungal testing methods, including challenges and optimization efforts. Finally, we improved the manuscript’s clarity by addressing the points you raised regarding concise discussions and limitations. Detailed and point-by-point responses are provided below, and the corresponding corrections are included in the resubmitted article, where you can see the changed sections highlighted in red.

2. Questions for General Evaluation

Reviewer’s Evaluation

Response and Revisions

Does the introduction provide sufficient background and include all relevant references?

Can be improved

Introduction has been improved

Are all the cited references relevant to the research?

Yes/Can be improved/Must be improved/Not applicable

Yes, all cited references have been carefully selected for their relevance to the research.

Is the research design appropriate?

Yes

Are the methods adequately described?

Yes

Are the results clearly presented?

Yes

Are the conclusions supported by the results?

Can be improved

The conclusion was improved by adding a future research plan, clarifying how the current work connects to upcoming studies and development directions.

Are all figures and tables clear and well-presented?

Yes

3. Point-by-point response to Comments and Suggestions for Authors

Comments 1: i) The authors do not provide sufficient characterization of the silicone support. It is unclear whether the substrate is crystalline or amorphous, which could significantly influence the surface activation process and coating efficiency. Clarifying the structural nature and relevant properties of the silicone material is essential for reproducibility and for understanding the observed results.

Response 1:

Thank you for your comment.

Based on information obtained from the silicone supplier, the substrates used in our study are amorphous Liquid Silicone Rubber (LSR). We have now added this characterization detail to the revised manuscript for clarity and reproducibility (Page 13; Line 443).

Comments 2: While the study discusses the limitations of traditional activation techniques, it does not include experimental comparisons with established approaches such as plasma activation or silanization. Incorporating such comparisons would strengthen the case for the proposed eco-friendly method by demonstrating its relative efficiency, stability, and safety.

Response 2: Thank you for this valuable suggestion. We agree that an experimental comparison with conventional methods could provide important insights into the relative efficiency, stability, and safety of our proposed method. However, conducting side-by-side experimental comparisons was beyond the current study's scope due to technical and resource limitations such as the absence of plasma equipment at our laboratory at this moment. Apart from the cost it would take considerable time for us to acquire such equipment.

The main aim of this study was to introduce a method capable of activating silicone surfaces without the need for specialized equipment such as plasma devices or the use of harmful chemicals typically involved in silanization making some of its advantages clear even without direct comparative studies.

Comments 3: The authors are encouraged to include direct experimental comparisons between the proposed activation method and conventional techniques, supported by quantitative data.

Response 3: We thank the reviewer for the suggestion. Our study focuses on demonstrating the effectiveness and safety of the proposed activation method for silicone surfaces. Direct quantitative comparisons with conventional techniques were not included due to technical limitations, such as restricted access to plasma equipment. We agree that such comparisons would be valuable, and once these limitations are resolved, we plan to optimize our method and conduct systematic, quantitative comparisons with established activation methods in future work.

Comments 4: The authors mention plans for future research but provide no preliminary data or discussion on potential challenges. Including initial stability and biocompatibility results (particularly under conditions mimicking real-world medical use) would enhance the practical relevance of the study.

Response 4: We thank the reviewer for this valuable suggestion. We agree that preliminary evaluations of stability and biocompatibility, especially under conditions simulating real medical use, are essential to demonstrate the practical relevance of our antibacterial coating. Our current study focused on developing and characterizing the chitosan-coated silicone surface, providing proof of concept and fundamental results. Preliminary experiments related to the stability of the coatings have been conducted, including tests of coated silicones in phosphate-buffered saline (PBS) at various pH levels and under sterilization conditions. Biocompatibility studies using cell models are also underway as part of our ongoing research program. However, including these detailed results was beyond the scope of this manuscript and will be reported comprehensively in a future publication, after allowing time to consider IP issues that may arise.

Comments 5: The antibacterial efficacy of chitosan-coated silicone is evaluated against four bacterial strains. Although these are clinically relevant pathogens, expanding the tests to include additional gram-positive, gram-negative, and fungal strains would provide a more comprehensive assessment of the coating’s antimicrobial spectrum. At minimum, a short comment addressing this limitation should be added, with an indication that broader studies could be presented in future work.

Response 5: Thank you for your suggestion. The antifungal activity of chitosan against a broad range of fungi, including Aspergillus niger, Penicillium pinophilum, Paecilomyces variotii, Trichoderma virens, and Chaetomium globosum, has been well documented in previous studies. Accordingly, we have already prepared these fungal species from DSMZ company and are currently preparing to evaluate the antifungal efficacy of the chitosan-coated silicone against them. However, quantifying fungal growth on silicone surfaces is challenging despite adherence to ISO 846 standard protocols for antifungal susceptibility testing. To improve sensitivity and accuracy beyond these standard methods, we are also investigating complementary approaches such as DNA staining and molecular quantification techniques. Optimization of these methods is ongoing and essential for robust and reliable evaluation of antifungal properties. Once fully optimized, comprehensive antifungal testing will be performed and included in future studies. This information has been added to the revised manuscript (Page 11; lines 380–391).

Comments 6: A concise discussion on long-term stability and biocompatibility under medically relevant conditions should be included to provide a more complete perspective on the method’s applicability.

Response 6: Thank you for your suggestion. A concise discussion on the long-term stability and biocompatibility of the chitosan coating under medically relevant conditions has now been included (Page 11; lines 386-397).

Comments 7: The authors disclose potential conflicts of interest related to a patent and a start-up company. While this transparency is commendable, independent validation of the findings would help mitigate concerns about bias and strengthen the study’s credibility.

Response 7: Thank you for addressing the concerns regarding potential conflict of interest and the validity of our research. To reduce potential bias and enhance transparency, we would like to emphasize that the method we presented was independently applied by several students involved in the project, using different silicone substrates. Although slight variations in results were observed between silicones from different suppliers, these differences can likely be attributed to variations in silicone diameter and surface properties. However, the overall outcome consistently demonstrated effective activation of all silicone types with our method. Additionally, the chitosan coating was successfully applied to various silicone substrates, and the antimicrobial effects were confirmed by different people working on this project. However, independent validation can only come from other research teams that have no connection to our team, and it is a part of the scientific process. The procedures we have been using are simple to perform, and all necessary information is fully disclosed, including all our data. The replication of this work should therefore be relatively straight forward for any outside researchers. We are committed to maintaining transparency and would welcome further independent replication to confirm our finding.

4. Response to Comments on the Quality of English Language

Reviewer’s Evaluation: The English is fine and does not require any improvement.

Response 1: Thank you for confirming that the language is satisfactory and does not need improvement.

Reviewer 3 Report

Comments and Suggestions for Authors

The submitted Manuscript describes production of antibacterial coating on silicone. The text is comprehensively written, all the methods are described in proper detail and the research design is clearly thought through. 

Dear authors, it was my pleasure to read this paper. Thank you for the great work. Here are some minor modifications that I suggest you to implement before publishing: 

1) Fig.1. I suggest writing the solvents and reactants used for the preparation of the samples right on the photographs. This will make it much easier to perceive. 

2) Lines 234-237: "The superior performance of toluene can be attributed to its ability to penetrate the silicone matrix, potentially increasing the available surface area for reaction. This effect was evidenced by a notable swelling of the silicone discs to an increased size when disctreated with toluene (Figure 2 A).  However, after subsequent washing with isopropanol and water, the disc size returned to its original dimensions." - Perhaps include images of the disk treated with toluene after washing with isopropanol and water in the Supplementary to prove that it does go back to its original size. 

3) Fig.2C. Please mention the solvent used in the preparation of this sample. Was it toluene?

4) As far as I can see Fig.3A is not referenced in the text, please fix that. 

5) Table 2. It would be nice to provide a more visual representation of the data presented in this table. Perhaps, a graph?

6) Correction of typos and some text formatting is required. I attach a pdf file where I highlighted some of the typos that I noticed, but there is certainly more. 

Author Response

Response to Reviewer 3 Comments

1. Summary

Dear Reviewer,

Thank you for your constructive suggestions and valuable feedback. In response to your comments, we have updated the annotation of Figure 1, added images of toluene-treated silicone disks before and after washing to the Supporting Information, corrected all references to images in the text, and edited for typographical and formatting errors.

All changes in the revised manuscript are highlighted in red for your convenience.

2. Questions for General Evaluation

Reviewer’s Evaluation

Response and Revisions

Does the introduction provide sufficient background and include all relevant references?

Yes

Are all the cited references relevant to the research?

Yes/Can be improved/Must be improved/Not applicable

Yes, all cited references have been carefully selected for their relevance to the research.

Is the research design appropriate?

Yes

Are the methods adequately described?

Yes

Are the results clearly presented?

Yes

Are the conclusions supported by the results?

Yes

Are all figures and tables clear and well-presented?

Can be improved

We have added solvents and reactants directly on Fig. 1 images and replaced Table 2 with a graph for better clarity.

3. Point-by-point response to Comments and Suggestions for Authors

Comments 1: Fig.1. I suggest writing the solvents and reactants used for the preparation of the samples right on the photographs. This will make it much easier to perceive.

Response 1:

Thank you for your suggestion. In the revised figure, the solvents and reactants used for sample preparation have been clearly labeled directly on the photographs (Figure 1, Page 5).

Comments 2: Lines 234-237: "The superior performance of toluene can be attributed to its ability to penetrate the silicone matrix, potentially increasing the available surface area for reaction. This effect was evidenced by a notable swelling of the silicone discs to an increased size when disctreated with toluene (Figure 2 A).  However, after subsequent washing with isopropanol and water, the disc size returned to its original dimensions." - Perhaps include images of the disk treated with toluene after washing with isopropanol and water in the Supplementary to prove that it does go back to its original size.

Response 2: Thank you for your suggestion. Images of the silicone disc treated with toluene after subsequent washing with isopropanol and water have been included in the Supplementary Material as recommended (Supplementary Material, Page 2; Figure S2).

Comments 3: Fig.2C. Please mention the solvent used in the preparation of this sample. Was it toluene?

Response 3: Thank you for your attention. The solvent used for the sample in Figure 2C was isopropanol. It has been mentioned in the modified figure caption in the revised manuscript (Page 7; Line 267).

Comments 4: As far as I can see Fig.3A is not referenced in the text, please fix that.

Response 4: Thank you for pointing this out. It is now corrected in the revised manuscript (Page 7; Line 276).

Comments 5: Table 2. It would be nice to provide a more visual representation of the data presented in this table. Perhaps, a graph?

Response 5: Thank you for your comment. This section has been thoroughly revised in the updated version of the manuscript. The data are now presented in graphical form for clearer visual interpretation, and the results are also expressed in more standard units (Page 12; Figure 7).

Comments 6: Correction of typos and some text formatting is required. I attach a pdf file where I highlighted some of the typos that I noticed, but there is certainly more.

Response 6: Thank you for highlighting the typos and formatting issues in your attached PDF. The manuscript has now been carefully reviewed, and all identified typos and formatting errors have been corrected.

4. Response to Comments on the Quality of English Language

Reviewer’s Evaluation: The English is fine and does not require any improvement.

Response 1: Thank you for confirming that the language is satisfactory and does not need improvement.

5. Additional clarifications

Reviewer’s comment regarding the quality of figures: Figures and tables can be improved

Response: Table 2 from the previous version of the manuscript has been revised, and its data are now presented in the form of a bar graph in Figure 7 for greater clarity in the revised version of the manuscript

Round 2

Reviewer 2 Report

Comments and Suggestions for Authors

The authors generally considered adequately to my observations. However, regarding my first comment the authors just answered that the used silicone substrates are amorphous platinum-cured Liquid Silicone Rubber (LSR). I consider that additional characterization is necessary for comprehensive reproducibility and scientific rigor. Specifically, the following details could enhance the manuscript:

i) Chemical Composition: Provide details on the exact formulation of the LSR, including any additives or fillers present.

ii) Surface Properties: Provide information on the initial surface roughness, hydrophobicity (contact angle measurements), and chemical functionality (e.g., FTIR analysis) of the untreated silicone.

iii) Supplier Specifications: Include detailed specifications from the silicone supplier, such as molecular weight, curing conditions, and degree of crosslinking.

iv) Microscopic Analysis: Use techniques like scanning electron microscopy or atomic force microscopy to analyze the surface morphology before and after activation.

These additional characterizations would provide a more complete understanding of the silicone material and its behavior during the activation and coating processes, ensuring reproducibility and enabling comparisons with other studies.

The authors must consider a major revision of their manuscript.

Author Response

Dear Reviewer,

Thank you very much for your time and for the constructive feedback on our manuscript.

As suggested, we have performed additional analyses on silicone before and after activation and coating, using contact angle measurements to determine hydrophobicity changes and scanning electron microscopy (SEM) to assess surface morphology, respectively. These data have been added to the Supplementary Information with a brief description in the manuscript to provide a clearer understanding of the surface modifications.​

We also attempted to investigate surface roughness using atomic force microscopy (AFM); however, we do not currently have access to AFM facilities in our faculty, and our attempts to collaborate with external laboratories have not yet been successful. Regarding your question about silicone preparation, we contacted the supplier and have included their response in this letter.​

Detailed point-by-point responses are provided below, and the corresponding corrections are included in the resubmitted article, where changed sections are highlighted in red.

Comment 1: Chemical Composition: Provide details on the exact formulation of the LSR, including any additives or fillers present.

Comment 3: Supplier Specifications: Include detailed specifications from the silicone supplier, such as molecular weight, curing conditions, and degree of crosslinking.

Response 1:

Thank you for your comments. In response to your comments 1 and 3, we contacted the silicone suppliers, and their reply is provided below for your reference.

“The information requested by the reviewer is mostly proprietary to either the supplier of the silicone or the manufacturer of the silicone samples. The aim of this study is to show that this method can be generally used for coating LSR silicones directly from a production line.  Here mechanical properties are used to characterize the cured silicone, and the different properties (durometer, tensile strength, tear strength, and compression set) must be within a range that is specified by the manufacturer. For the current samples, the durometer is e.g., expected to be in a range of 35-45 on the Shore OO scale.  A typical LSR silicone blend is a statistical mix of different chain lengths and therefore molecular weight, and level of crosslinking are not well defined.  Curing conditions are already reported in the manuscript. Having to go to molecular level in defining the material used, would render this approach very difficult to use in manufacturing conditions, which is the ultimate goal of this approach.”

 Comments 2: Surface Properties: Provide information on the initial surface roughness, hydrophobicity (contact angle measurements), and chemical functionality (e.g., FTIR analysis) of the untreated silicone.

Response 2: Thank you for this helpful comment. Information on the surface properties of the untreated silicone has now been clarified in the Supporting Information. We have performed water contact angle measurements to investigate the wettability of the silicone samples, and the results have been added to the Supporting Information (Figure S3; Page 4) with brief description in the revised manuscript (Page 11, Lines 351 – 354; Page 14, Lines 473 – 476). We also attempted to investigate surface roughness using atomic force microscopy (AFM); however, we do not currently have access to AFM facilities in our faculty, and our attempts to collaborate with external laboratories have not yet been successful. In addition, FTIR measurements for untreated silicone, and chitosan‑coated silicone were already included in the manuscript; these spectra are presented in Figure 5 (Page 10)

Comments 4: Microscopic Analysis: Use techniques like scanning electron microscopy or atomic force microscopy to analyze the surface morphology before and after activation.

Response 4: Thank you for your comment. We performed scanning electron microscopy (SEM) analysis on the untreated silicone, activated silicone, and chitosan-coated silicone. The results are provided in the Supplementary Information (Page 4; Figure S4) with a brief description in revised manuscript (Page 11, Lines 354 – 358; Page 14, Lines 473 – 476).

Round 3

Reviewer 2 Report

Comments and Suggestions for Authors

The authors have improved the quality of their manuscript, which is now ready to be published.